# Surgical Management of Lower Back Pain: Is Optimizing Spinopelvic Alignment Beneficial for Patient Outcomes?

**DOI:** 10.3390/life15060833

**Published:** 2025-05-22

**Authors:** Dieter Thijs, Ardavan Kashtiara, Sarah Beldé, Erik Van de Kelft

**Affiliations:** 1Department of Neurosurgery, Antwerp University Hospital, Drie Eikenstraat 655, 2650 Edegem, Belgium; ardavan.kashtiara@vitaz.be; 2Faculty of Medicine, and Health Sciences, University of Antwerp, Universiteitsplein 1, 2600 Wilrijk, Belgium; sarah.belde@hotmail.com (S.B.); erik.vandekelft@vitaz.be (E.V.d.K.); 3Department of Neurosurgery, Vitaz, Moerlandstraat 1, 9100 Sint-Niklaas, Belgium

**Keywords:** sagittal balance, spinopelvic parameters, degenerative lumbar spine disorder, spinal arthrodesis, spinal surgery, clinical outcomes, persistent lower back pain, adjacent segment disease

## Abstract

For the past two decades, the consideration of spinopelvic parameters, sagittal balance, and spine shape has gained importance in the diagnosis and optimal surgical management of painful adult spinal deformity. These principles are used with increasing frequency in the surgical planning and treatment of degenerative mechanical lower back pain. Several parameters exist to analyze both global and regional spinal balance. Chronic lower back pain due to degenerative disc disease, degenerative spondylolisthesis, or adult spinal deformity can be surgically managed in a multitude of ways ranging from simple decompression to multilevel arthrodesis with or without corrective osteotomies, depending on the presumed etiology of the pain, surgical planning, and the surgical goal. In surgical candidates, preoperative evaluation of spinopelvic parameters is paramount, as increasing evidence shows that restoration of the shape of the spine while respecting these parameters improves patient-reported outcome measures (PROMs), decreases re-operation rates, and reduces mechanical complications such as proximal junctional kyphosis/failure (PJK/PJF), distal junctional kyphosis/failure (DJK/DJF), adjacent segment disease (ASD), and rod fracture. This review provides a conceptual analysis of spinopelvic alignment, global and regional sagittal balance, and the restoration of the spine’s shape in relation to patient outcomes during surgical treatment of degenerative spine disorders.

## 1. Introduction

Degenerative lumbar spine disorders are a prevalent cause of chronic lower back pain (cLBP) and have a significant socio-economic burden [1]. The French Society of Spinal Surgery classifies cLBP into three categories: non-degenerative cLBP, degenerative cLBP, and cLBP of unknown origin. Non-degenerative cLBP is defined as pain resulting from trauma, congenital spondylolysis, tumor, infection, or an inflammatory process. Degenerative cLBP, previously known as non-specific cLBP, is characterized by varying combinations of degenerative changes, with or without regional and/or global alterations in spinal alignment. The pain in cLBP of unknown origin does not relate to anatomical abnormalities [2]. This paper focuses on degenerative cLBP.

Potential sources of degenerative cLBP include the intervertebral discs, facet joints, vertebral endplates, vertebral bodies, spinal ligaments, and paraspinal muscles, or a combination thereof. Additionally, regional spinal disorders—such as degenerative spondylolisthesis—and global malalignment, including loss of sagittal or coronal balance, may contribute to mechanical cLBP [2,3]. Degenerative changes that may cause neural compression—such as disc herniation, foraminal stenosis, or spinal canal stenosis—can result in radicular pain and/or neurogenic claudication in addition to cLBP. These conditions, however, are beyond the scope of this paper.

Though various studies have been performed, no conclusive evidence has been demonstrated to support either conservative treatment (e.g., physiotherapy, medication, multimodal and interventional pain management) or surgical interventions for degenerative cLBP [3]. Clinical studies frequently report postoperative complications, residual mechanical pain, adjacent segment disease (ASD), and proximal junction kyphosis.

Spinal surgery generally has one of two main goals: decompress neural structures or restore spinal balance [4]. Over the past two decades, the importance of spinopelvic parameters in planning and performing spinal instrumentation has gained increasing importance. As a result, spine surgeons are now more frequently applying these principles in the treatment of degenerative lumbar spine disorders. The hypothesis is that aligning lumbar fixation with a patient’s individual spinopelvic parameters can improve clinical outcomes by reducing persistent mechanical cLBP and lowering the risk of adjacent segment disease (ASD) [5,6]. Despite this, there is still ongoing debate within the spinal surgery community about whether spinopelvic parameters should be routinely considered when planning lumbar arthrodesis, whether for short or long constructs.

This review aims to examine whether the regional and global spinal balance is beneficial for patients with mechanical cLBP caused by degenerative disorders who are undergoing spinal surgery. It also explores the influence of spinopelvic parameters on postoperative outcomes. Fore readers who may not be familiar with these parameters, a brief summary and illustration will be included to support understanding of the concepts discussed in this review.

### 1.1. Spinopelvic Parameters

Spinopelvic parameters define the position of the sacrum within the pelvis (static parameter) and the degree of pelvic retroversion (dynamic parameters) required to maintain ergonomic spinal sagittal balance [7]. The parameters discussed in this article are explained below.

### 1.2. Pelvic Incidence

In 1998, Jean Legaye, a Belgian orthopedic surgeon, along with his co-authors, proposed an anatomical parameter that plays a key role in determining spinal balance [8]. The pelvic incidence (PI) was defined as the angle between the perpendicular to the sacral plate at its midpoint and a line connecting this point to the hip axis (the center of the bicoxofemoral axis). See Figure 1a. PI is a morphological parameter defining the position of the sacrum relative to the femoral heads. It is important to note that this is the only non-positional (i.e., static) parameter. Consequently, patients cannot alter their PI by changing positions. PI is considered specific for each individual and is often referred to as an anatomical parameter, remaining unchanged after skeletal maturity [8]. The normal value for PI is 53° ± 9°, but there is considerable variability, as each individual has their own ‘normal’ PI [9]. PI is related to the ability to compensate for sagittal imbalance, with higher PI values indicating a greater capacity for sagittal compensation [10]. This is because PI, as described below, is the sum of pelvic tilt (PT) and sacral slope (SS), allowing for increased pelvic retroversion (i.e., higher PT) in individuals with higher PI. PI is associated with specific spinal pathologies; for instance, a higher PI is linked to the development of spondylolisthesis due to increased shear stress, which facilitates disc slippage [11,12].

1.Pelvic tilt

Pelvic tilt (PT) is a dynamic, positional parameter that is defined as the angle between a vertical line passing through the midpoint of the femoral heads and a line connecting this point to the center of the sacral plate [10] (see Figure 1b). As a positional parameter, PT changes with pelvic orientation, particularly its rotation around the acetabula. PT increases during pelvic retroversion, acting as one of the body’s compensatory mechanisms to maintain sagittal balance [13,14]. Normal values for PT range around 13° ± 6° [13]. Because PT is a component of PI, it can be more accurately estimated using the following formula: PT = 0.44 × PI − 11° [15]. For practical use, a simplified version of the formula can be applied: PT = PI/2 − 11°.

2.Sacral slope

Sacral slope (SS) is the second dynamic parameter and is defined as the angle between the horizontal plane and the sacral endplate [10], as shown in Figure 1c. Typical values for SS are 41° ± 8°. Among all spinopelvic parameters, SS shows the strongest correlation with spinal lordosis—though not necessarily lumbar lordosis—and the overall sagittal shape of the spine, as described in the Roussouly classification [15,16,17,18].

### 1.3. Lumbar Lordosis

Lumbar lordosis (LL) refers to the angle formed between the upper endplate of the first lumbar vertebra (L1) and the sacral endplate [11] (Figure 2a). Research has shown that individuals with low LL and SS are more likely to experience lower back pain [19]. The normal range for LL is between 30° and 79° [11]. Because LL is directly correlated with SS, which in turn depends on PI, the formula derived from an asymptomatic population is LL = PI/2 + 28° [15].

### 1.4. Spinal Lordosis

Spinal lordosis refers to the anatomical curvature of the lumbar spine. Spinal lordosis is influenced by the shape of the lumbar and thoracic vertebrae as well as the presence or absence of ribs. This curvature begins at the sacrum and extends upward to the point where thoracic kyphosis begins—a transitional point known as the inflection point (Figure 2b). Contrary to the strict anatomical definition, the inflection point does not always correspond with the Th12-L1 vertebrae but can vary between Th10 and L2, depending on the spinal morphology [20]. Spinal lordosis can be divided into two arcs: the lower arc of lordosis (LAL), which extends from S1 to the apex of the curve, and the upper arc of lordosis (UAL), which extends from the apex to the inflection point [21]. The UAL typically measures around 20° and is correlated with the lower arc of thoracic kyphosis. The LAL is approximately equivalent to the SS, with greatest curvature generally occurring between L4 and S1 [20]. In healthy individuals, the position of the lordotic apex is also linked to PI, with a lower apex associated with a lower PI, and a higher apex with a higher PI [22]. These two arcs can be combined to calculate the spinal lordosis ratio (SLR), a metric that corresponds to the different Roussouly types and serves as a key tool for analyzing and restoring sagittal alignment [22].

In surgical planning, the goal is to achieve a lumbar lordosis angle that closely corresponds to the patient’s PI. It is further recommended that approximately 60% of this correction be achieved in the lower arc, from L4 to S1, while ensuring that the reconstructed apex is positioned at a height appropriate to the individual’s PI.

### 1.5. Sagittal Vertical Axis

The sagittal vertical axis (SVA) is defined as the horizontal distance between the C7 plumb line and the posterior superior corner on the upper sacral plate [10,23] (Figure 3a). It quantifies the global sagittal alignment of the spine in relation to the pelvis. Typical SVA values are 0.5 cm ± 2.5 cm. [24,25], and an SVA of less than 5 cm is generally considered an acceptable surgical target [26]. Unlike angular parameters, SVA is an absolute linear measurement and can vary significantly between individuals with identical angular deviations, particularly if they differ in height. As a result, calibrated radiographic imaging is essential for accurate assessment of SVA.

### 1.6. Odontoid-Hip Axis

The odontoid-hip axis (OD-HA) is a global sagittal alignment parameter that evaluates the alignment of the entire spine and pelvis, excluding the skull (see Figure 3b). It is defined as the angle between a line connecting the center of the odontoid process (C2) to the center of the femoral heads and the vertical axis passing through the center of the femoral head. This angular measurement is often preferred for assessing overall spinal balance, with typical values ranging from +3° to −5° [25]. Ferrero et al. described an abnormal OD-HA as any value exceeding 6.1° (>2 standard deviations from the mean) and found it to be associated with increased disability [27].

### 1.7. T1 Pelvic Angle

The T1 pelvic angle (T1PA) is a radiographic parameter used to assess global malalignment and compensatory pelvic retroversion. It is measured as the angle between the line from the bicoxofemoral head axis to the center of T1 and the line from the same axis to the midpoint of the S1 endplate. In a study involving 559 patients with spinal deformity, a strong correlation was found between the T1PA and the Oswestry Disability Index (ODI). T1PA distributions (<10°, 10–20°, 21–30°, and >30°) were associated with progressively worsening health-related quality of life [28].

### 1.8. Other Measurements

A variety of additional measurements, ratios, and scores have been described in the literature. While the parameters discussed above are among the most widely used and well-known, several other metrics can also be valuable in clinical practice. These include the Global Alignment and Proportion score, spinosacral angle (SSA), cervical inclination angle (CIA), Barrey ratio, global tilt angle (GTA), and lumbar lordosis index (LLI) [29,30,31,32,33].

## 2. Materials and Methods

A search was conducted on PubMed using the following terms: “sagittal balance”, “spinopelvic parameters”, “lumbar degenerative disc disease”, “lumbar degenerative spondylolisthesis”, “lumbar degenerative disc herniation”, “lumbar degenerative spinal stenosis”, “lumbar degenerative scoliosis”, and “clinical outcome”. Various combinations of these terms were used with AND/OR operators to diversify the search, resulting in the identification of 2559 records. After removing duplicates, titles and abstracts were screened. Records were excluded based on population criteria (as outlined above), study design (only studies comparing clinical outcomes between well-balanced and imbalanced groups were included), and language (only articles available in English were considered). Finally, the full texts were assessed for eligibility. Studies were excluded if they did not report spinopelvic parameters or failed to examine their association with clinical outcomes between well-balanced and imbalanced groups. Furthermore, a minimum of 30 study participants and six months of follow-up was required. In total, 19 articles met the criteria and were included in this review, composed of 3 prospective and 16 retrospective cohort studies. 

## 3. Results

### 3.1. Global Disorders of Coronal/Sagittal Balance

In the only prospective study investigating mechanical cLBP due to degenerative lumbar spinal disorders, Lafage et al. reported the results for 33 patients. They found that a high postoperative PT, serving as a compensatory mechanism for sagittal imbalance, was significantly correlated with poorer clinical outcomes and reduced quality of life (*p* < 0.001) [34]. In terms of ODI, the strongest correlation was observed with walking disability (*p* = 0.001). When the study population was subdivided into five equally sized groups based on increasing PT, this study showed a clear trend: greater PT was linked to more pain and reduced function. Key findings include that self-reported disability increases with sagittal imbalance, the severity of disability worsens with pelvic retroversion, and pelvic parameters should be considered in sagittal plane analysis [34].

Recently, Park et al. published a retrospective study involving 156 patients with degenerative lumbar scoliosis treated with posterior instrumentation. Patients were divided into two groups: the compensatory balanced group (i.e., SVA < 5 cm) and the imbalanced group (i.e., SVA ≥ 5 cm). At the final follow-up (mean duration: 50 months), patients in the imbalanced group had significantly worse ODI and Scoliosis Research Society-22 (SRS-22) scores (*p* < 0.01) [35]. These findings suggest that patients with a persistent forward-leaning spine following thoracolumbar balance correction experience inferior clinical outcomes compared to those with good sagittal balance (*p* < 0.01).

In a retrospective cohort study involving the same study population, Zhou et al. examined 71 patients [36]. They proposed an equation to estimate the ideal PI-LL to be 0.52 × age + 0.38 × PI − 39.4 (*p* = 0.001) [36].

In an earlier study involving 130 patients from the same population, Zhou et al. found that the postoperative LL-TK was strongly correlated with quality of life and SVA [37]. The threshold for achieving a good clinical outcome of ODI < 20 and good sagittal balance (SVA < 5 cm) was set at 10° for LL-TK. Patients with LL-TK > 10° had significantly better postoperative VAS, ODI, and JOA scores (*p* < 0.01) [38].

Zhang et al. conducted another retrospective analysis of 44 patients who underwent posterior spinal instrumentation for degenerative lumbar scoliosis [39]. They reported that patients with postoperative PI-LL between 11 and 20° had lower ODI scores compared to those with PI-LL < 11° (17.3 ± 4.9 vs. 26.0 ± 5.4, *p* < 0.05) and PI-LL > 20° (17.3 ± 4.9 vs. 32.4 ± 7.3, *p* < 0.05). A Pearson correlation analysis further confirmed that clinical outcome scores were negatively associated with high PI-LL values [39]. They concluded that the optimal PI-LL value may be achieved between 10° and 20° with good clinical outcomes.

Sun et al. retrospectively examined 74 patients who underwent posterior spinal instrumentation for degenerative lumbar spinal disorders [37]. They reported a positive correlation between lower PI-LL scores and higher final ODI scores (*p* < 0.001). There was also a significant difference in adjacent segment disease between the higher PI-LL group and the ideal PI-LL group (*p* = 0.038) [40]. They concluded that the optimal PI-LL value may be achieved between 10° and 20° with excellent clinical outcomes and a lower ASD.

Finally, Simon et al. retrospectively evaluated 47 patients who received posterior pedicle screw fixation for degenerative lumbar spinal disorder [41]. They reported a significant correlation between the SF-36 and the postoperative SS (*p* = 0.04) and LL (*p* = 0.0491) [41]. Their findings emphasize the importance of restoring sagittal balance to improve functional outcomes and overall quality of life in patients with combined coronal and sagittal imbalance [41].

### 3.2. Regional Disorders of Spinal Balance/Alignment

Bourghli et al. [40] prospectively followed 30 patients with degenerative lumbar spondylolisthesis who underwent posterior lumbar interbody arthrodesis (PLIA) [42]. They observed that the four patients with poor outcomes (i.e., ODI improvement of less than 25%) had the same or increased SVA and decreased LL as compared to preoperative measurements [40].

In another prospective cohort study, Lenz et al. followed 32 patients with degenerative lumbar spondylolisthesis who underwent PLIA. A significant correlation (*p* = 0.041) was observed between restoration of SS and favorable clinical outcome scores [43].

Recently, Tan et al. retrospectively analyzed 303 patients with degenerative lumbar spinal disorders [42]. This population consisted of spondylolisthesis, spinal stenosis, and disc herniation patients who underwent PLIA. The PI-LL match group (PI–LL ≤ 10°) showed significantly greater VAS improvement rates (79.50% ± 24.74% vs. 69.15% ± 33.39%, *p* = 0.003) and ODI improvement rates (68.93% ± 40.69% vs. 56.83% ± 55.33%, *p* = 0.034) compared to the PI-LL mismatch group (PI–LL > 10°) in the single-level arthrodesis group. However, no differences in clinical outcomes were observed between the two level groups in the two- or three-level arthrodesis or in the single L3/L4 arthrodesis [42]. 

Thornley et al. retrospectively analyzed 243 patients with degenerative lumbar spondylolisthesis who underwent decompression with or without PLIA [44]. A PI-LL mismatch (PI–LL > 10°) was correlated with more severe disability (ODI, 0.134, *p* < 0.05), greater leg pain (0.143, *p* < 0.05), and significantly more back pain (0.189, *p* < 0.001). A reduction in LL was linked to higher disability (*p* = 0.027) and significantly more back pain (*p* < 0.001). Additionally, higher SVA values were correlated with poorer patient-reported functional outcomes. An increase in SVA was associated with more NRS back pain (*p* < 0.001) and higher NRS leg pain scores (*p* = 0.046), regardless of surgery type [44].

He et al. observed a significant reduction in PT (*p* < 0.001) in a non-persistent back pain group (−3.7 ± 5.3) compared to a persistent back pain group (+2.2 ± 4.2), while retrospectively examining 107 patients with degenerative lumbar spondylolisthesis undergoing PLIA [45]. They observed a significantly lower PT difference pre- and postop between the persistent back pain and non-persistent back pain groups (2.2 ± 4.2° vs. −3.7 ± 5.3°, *p* < 0.001). Additionally, the persistent back pain group demonstrated a significantly smaller change in segmental lordosis (0.5 ± 4.3° vs. 3.8 ± 5.1°, *p* = 0.02). They concluded that segmental lordosis, disc height, and PT improved significantly less in the cLBP group compared to non-cLBP (*p* < 0.05) [45].

Another retrospective analysis of 120 patients with degenerative lumbar spondylolisthesis and spinal stenosis found significantly lower PT in the non-persistent cLBP group (15.1° ± 7.3° vs. 22.3° ± 10.8°, *p* = 0.0001). Furthermore, significantly higher LL was observed in the non-persistent back pain group (42.2° ± 11.2° vs. 35.8° ± 8.7°, *p* = 0.027). Additionally, there was a significant improvement regarding PI-LL mismatch in the non-persistent back pain group (*p* = 0.006). Finally, a significant reduction in the SVA was observed in both groups (*p* < 0.05), though the reduction was greater in the non-persistent back pain group (*p* < 0.001). The authors concluded that patients with decreased PT, improved SVA, and PI-LL match (PI-LL < 10°) experienced less cLBP (*p* < 0.05) [46].

Another 63 patients with degenerative lumbar spondylolisthesis were retrospectively analyzed by Liow et al. [47] after undergoing PLIA. They observed that patients with postoperative SS ≥ 30° had significantly lower back pain (*p* < 0.04). While there were no differences in leg pain or outcome scores, there was a trend towards better outcomes and higher satisfaction/expectation fulfilment in patients with postoperative SS ≥ 30. The area under the curve for the change in SS was 0.680 (95% confidence interval, 0.453–0.907) for predicting favorable clinical outcomes. They concluded that patients with increased SS (≥30 degrees) experienced less back pain after short-segment lumbar arthrodesis, which was associated with increased LL postoperatively [47].

Sun et al. retrospectively analyzed 163 patients with degenerative lumbar spinal disorder who underwent PLIA [48]. Logistic regression analysis and receiver operating characteristic analysis confirmed that preoperative PT > 24.1° and thoracic TK > 23.3° were significant risk factors of adjacent segment disease (*p* < 0.05). In their conclusion, they stated that PI-LL values between 10° and 20° were associated with excellent clinical outcomes and a reduced rate of adjacent segment disease (*p* < 0.05) [48].

Finally, Aoki et al. examined 52 patients with degenerative lumbar spinal disorder who underwent PLIA. They reported that a greater PI-LL mismatch significantly corresponded to lower postoperative VAS scores for cLBP (*p* = 0.045) and lower extremity pain (*p* = 0.024) [49].

Further retrospective studies analyzing patients with degenerative lumbar spinal disorder undergoing posterior spinal instrumentation identified PT, SS, SVA, and PT as risk factors for persistent cLBP and adjacent segment disease [50,51,52].

### 3.3. Evidence Table

Table 1 provides an overview of the selected articles, ranked by the level of evidence.

## 4. Discussion

In this article, we reviewed studies supporting the use of spinopelvic parameters in the surgical management—specifically through instrumentation—of regional and global spinal alignment disorders. However, the overall quality of evidence remains limited, primarily due to the retrospective nature of most studies. The available data suggest that both clinical and radiographic outcomes are superior in patients whose spinopelvic parameters are appropriately restored compared to those undergoing similar procedures without consideration of these parameters. Although the current evidence is limited, it does indicate that addressing spinopelvic alignment during surgery may lead to better clinical outcomes in the treatment of degenerative spinal disorders. Failing to consider these parameters may negatively impact patient outcomes, as suggested by the existing literature. 

In general, the analysis of postoperative results for cLBP surgery poses considerable challenges in generating high-quality qualitative evidence. Study populations are often highly heterogeneous, and the lack of randomization introduces a significant risk of selection bias. Moreover, the underlying source of spinal pain is frequently unclear or inadequately defined. For example, the omission of mechanical lower back pain in reporting contributes to selective reporting bias in many studies. Despite ongoing research efforts, a substantial knowledge gap remains in the accurate diagnosis of cLBP. Retrospective comparisons often involve dissimilar patient groups—effectively comparing apples to oranges. Additionally, variability in surgical techniques and surgeon expertise further complicates outcome interpretation, although these factors likely play a significant role.

A primary focus of outcome assessment is often pain—an inherently subjective experience influenced by biological, psychological, and social factors—which presents significant challenges in quantification. It is essential to differentiate between pain types, particularly mechanical, myofascial, neuropathic, and inflammatory pain, as only patients with mechanical or specific lower back pain, with or without neurological compromise, are likely to benefit from surgical intervention. Without a clear understanding of the underlying pain mechanism or accurate identification of the pain generator, surgery should not be considered. Instead, a thorough clinical assessment is essential, after which imaging studies may be used to support the diagnosis. However, it is important to recognize that radiographic evidence of lumbar pathology is frequently present in asymptomatic individuals, which may lead to diagnostic and therapeutic misinterpretation [53].

Incorporating spinopelvic parameters as additional radiographic measurements may enhance our understanding of cLBP in patients undergoing surgical fixation. These parameters provide several objective measures that can assist in diagnosis, surgical planning, intraoperative decision-making, and postoperative evaluation. At the very least, postoperative radiographs offer a quantifiable means to assess whether surgical correction has been adequately executed, and the availability of objective parameters enhances our understanding of the clinical and radiographic aspects we are measuring.

Schwab et al. proposed an ideal lumbar lordosis within the range of PI ± 9° and recommended a surgical target of a PI-LL mismatch of less than 10°. Failure to achieve this mismatch has been linked to poorer clinical outcomes [26,34]. Additionally, because lumbar lordosis needs to adapt to pelvic morphology, a mismatch between the two has been shown to prevent the patient from finding a posture that keeps the spine well-aligned [10].

Regarding the aging spine, the formula proposed by Zhou et al. appears to be well-founded. Their research suggests that less aggressive correction may be more appropriate in older patients, with a lower target LL proving sufficient to achieve acceptable clinical outcomes [36]. Our center is currently evaluating the role of PI in determining the ideal PI-LL mismatch. Preliminary data indicate that a greater PI-LL mismatch may be beneficial as PI increases, suggesting that less aggressive correction is warranted for individuals with a higher PI. We plan to further explore and publish these findings in the future.

Several important considerations should be noted regarding the interpretation of parameters in relevant studies. Many studies show a reliance on surrogate markers for global alignment and attempt to establish correlations with clinical outcomes. It is essential to recognize that the primary goal of sagittal balance is to achieve a horizontal gaze while maintaining posture within the “cone of economy”. This requires that both the spine above C7 and the head be included in the analysis of global spinal alignment. However, few of the studies analyzed in this research address this, often overlooking the cervical spine’s compensatory mechanisms and head positioning. Key measurements, such as the odontoid-hip axis (ODHA/C2HA), the center of the auditory meatus-hip axis (CAM-HA), the C0–C2 angle, and the chin–brow angle, can be derived from full-spine radiographs, allowing for a more comprehensive analysis of global alignment [15,54].

On the caudal side of the spine, the lower limbs play a significant role in global sagittal alignment. Knee flexion and ankle dorsiflexion serve as important compensatory mechanisms for the loss of sagittal balance. An initial assessment can be conducted during clinical examination, but precise measurements can only be made using full-body radiographs. From these radiographs, parameters such as the femoral obliquity angle and the full balanced index (FBI) can be extrapolated to provide a more detailed analysis of sagittal alignment [15,54,55,56].

While the SVA and PI-LL are widely used and have their merits, there are important considerations concerning their interpretation. The SVA is measured in absolute distance, in a world of angles. The advantage of angles, as stated earlier, is their applicability on non-calibrated images and the relativity of their value, which is independent of the height or width of a patient. Neither the SVA nor PI-LL consider locoregional balance, as one may have a severe locoregional balance problem but normal SVA and PI-LL values.

Thanks to Stagnara’s work in 1982 and the work of many others, we know that spinal lordosis is not always an anatomical lumbar lordosis [20,21,57,58]. Every spine, dictated by the SS and PI, has a certain shape, a given lordosis, apex, inflection point, and kyphosis. Spinal lordosis includes all lordotic vertebrae, which can number more or less than five. As previously mentioned, the apex divides the lordotic vertebrae into two arcs, creating an upper (UAL) and a lower (LAL) arc. The UAL was found to be constant among all spinal morphotypes around 20° ± 5° and equals the lower arc of adjacent kyphosis (not ‘thoracic kyphosis’ per se, since kyphosis can start at L2) [20,22]. These relationships are of great importance when analyzing radiographs.

Chevillotte et al. described the spinal lordosis ratio (SLR) in 373 healthy volunteers. This ratio is different for all five Roussouly types, with a strong and linear correlation to SS, and shows that lordosis is defined by the length (i.e., the number of vertebrae) and distribution among the different segments [22]. Disrespecting this ratio, may create segmental imbalance and a cascade of events leading to ASD, dissatisfactory outcomes, or mechanical failure (e.g., proximal junctional kyphosis). Neither PI-LL nor SVA account for SLR, which proves to be an inherent problem when interpreting results where these parameters are used.

While results often report on PT, other mechanisms by which patients attempt to restore balance, the so-called compensatory mechanisms, are often not mentioned. They include craniocervical collision (decrease in the C0–C2 angle), cervical hypo- or hyperlordosis (e.g., in PJK), thoracic hyperextension/lordosis, segmental retrolisthesis or hyperlordosis (increasing discal shear stress), and the aforementioned lower limb actions. Although more labor-intensive, including these outcomes in future publications would provide a more comprehensive understanding and could offer valuable insights for the future of surgical practice.

## 5. Conclusions

Fixation of the (thoraco-)lumbar spine as a surgical treatment of cLBP due to degenerative lumbar spinal disorders without considering the restoration of sagittal balance as indicated by the patient-specific spinopelvic parameters seems to be associated with poorer patient outcomes. While not all the pathophysiological mechanisms of pain are easily understood, it is sensible to strive for and restore physiological sagittal balance, since low-grade evidence supports this approach (Grade C recommendation). Previous research has contributed valuable insights to our community, ranging from basic biomechanics to studies involving healthy volunteers, which enhance our understanding of this complex condition. Treating degenerative spine disorders, whether through single-level decompression or multilevel thoracolumbar reconstruction, can be challenging. However, understanding and following data-driven principles may make this challenge more manageable, which will be rewarding for surgeons and patients in both the short and long term.

## Figures and Tables

**Figure 1 life-15-00833-f001:**
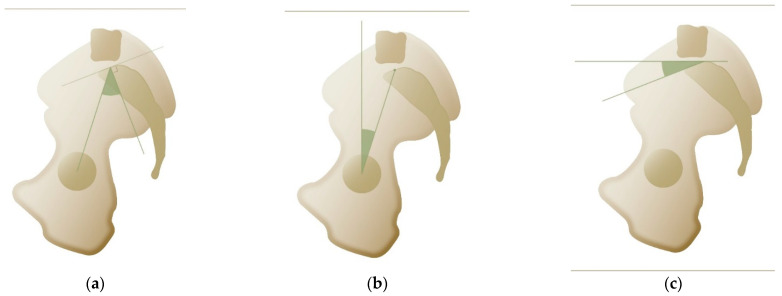
(**a**) Pelvic incidence (green angle). (**b**) Pelvic tilt (green angle). (**c**) Sacral slope (green angle).

**Figure 2 life-15-00833-f002:**
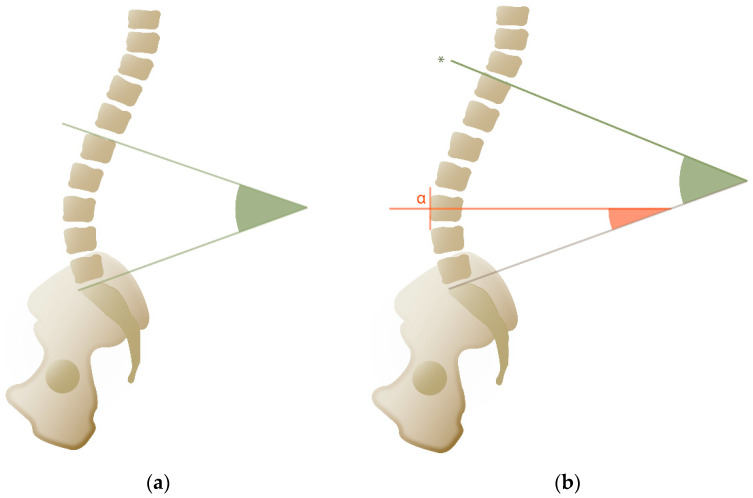
(**a**) Lumbar lordosis (green angle). (**b**) Spinal lordosis (green angle), lower lordosis (orange angle), apex (α), and inflection point (*).

**Figure 3 life-15-00833-f003:**
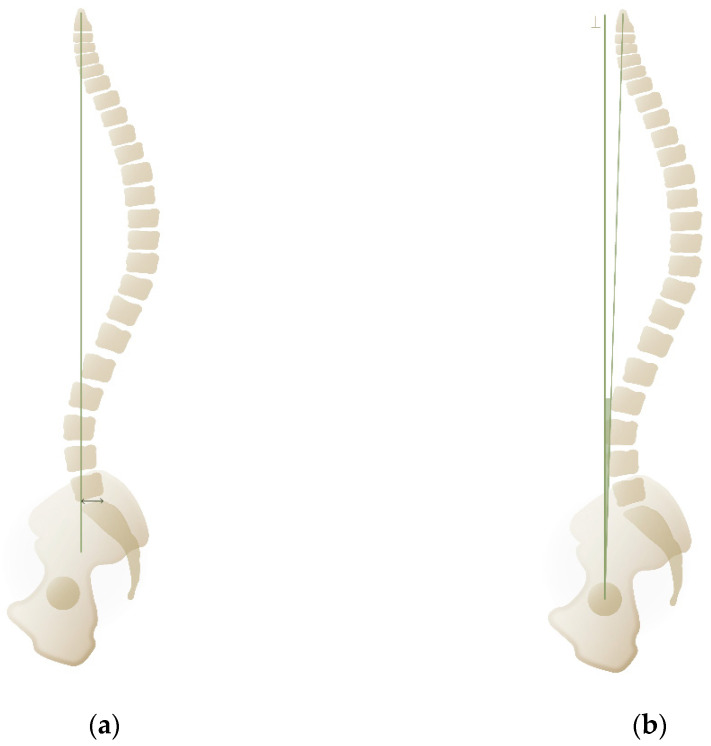
(**a**) Sagittal vertical axis (arrowline). (**b**) Odontoid hip axis (green angle).

**Table 1 life-15-00833-t001:** Overview of current clinical trials describing the relationship between sagittal imbalance correction and clinical outcomes for degenerative lumbar spine disorder.

Study	Population	Intervention	Follow-Up	Outcome	Method	Spinopelvic Parameters	Conclusion	Level of Evidence
**Prospective observational studies**								
Lenz, 2022 [43]	Spondylolisthesis *n* = 32	PLIA 1 or 2 levels	3 years	ODI EQ-5D	Multivariate analysis	SS Sagittal rotation Sacral inclination	Restoration of sagittal parameters showed significant correlation with improvement in clinical outcomes.	Level III
Bourghli, 2011 [40]	Spondylolisthesis *n* = 30	PLIA	Mean 36 months	ODI	Multivariate analysis	SVA	All patients with poor clinical outcomes had the same or increased SVA.	Level III
Lafage, 2009 [34]	Spinal stenosis *n* = 33	Posterior spinal instrumentation	Unknown	ODI SF-12 SRS	Multivariate analysis	PT	High PT is a compensatory mechanism for sagittal imbalance and was correlated with poor clinical outcomes and quality of life.	Level III
**Retrospective cohort studies**								
Park, 2024 [35]	Scoliosis/sagittal imbalance *n* = 156	Posterior spinal instrumentation 5 levels or greater	Mean 50 months	ODI SRS-22	Retrospective SVA < 5 cm SVA ≥ 5 cm	SVA	The group with suboptimal correction of SVA had significantly inferior clinical outcomes compared to the compensatory balance group.	Level IV
Zhou, 2023 [36]	Scoliosis/sagittal imbalance *n* = 71	Posterior spinal instrumentation	2 years or more	ODI SRS-22 HRQOL	Retrospective PI-LL < 10° PI-LL = 10–20° PI-LL > 20°	PI-LL LL SVA	PI-LL is an important spinopelvic parameter that affects the postoperative HRQOL. LL should be restored during surgery with the aim of achieving LL ≥ PI-14°. As age increases, lower LL goals are necessary for age-adjusted correction.	Level IV
Tan, 2023 [42]	Spondylolisthesis Spinal stenosis Disc herniation *n* = 303 patients	PLIA	2 years or more	ODI VAS	Retrospective PI-LL 10° or less PI-LL > 10	PI-LL	Postoperative PI-LL has an effect of patient’s quality of life and degree of lower back pain.	Level IV
Thornley, 2023 [44]	Spondylolisthesis *n* = 243	Decompression +/− interbody arthrodesis	1 year	ODI VAS	Retrospective Multivariate analysis	PI-LL LL SVA	Global postoperative spinopelvic parameters are correlated with functional outcomes.	Level IV
He, 2020 [45]	Spondylolisthesis *n* = 107	PLIA 1 or 2 level	6 months	ODI VAS	Retrospective PLBP Non-PLBP	PT SL	SL, height of disc, and PT were significantly less improved in the PLBP group compared to non-PLBP.	Level IV
Li, 2020 [46]	Spondylolisthesis Spinal stenosis *n* = 120	OLIA 1 level	2 years or more	ODI VAS	Retrospective PLBP Non-PLBP	PI-LL PT SVA	High PT, PI-LL mismatch, and high SVA were identified as significant risk factors for PLBP. Decreased PT, improved SVA, and improved PI-LL were associated with better functional outcomes.	Level IV
Liow, 2020 [47]	Spondylolisthesis *n* = 63	PLIA L4/L5	2 years	ODI Short-form 36	Retrospective SS ≥ 30° SS < 30°	PT PI LL SS	Patients with increased SS (≥30 degrees) experienced less back pain after short-segment lumbar arthrodesis surgery. This was associated with increased LL postoperatively, indicating better sagittal balance.	Level IV
Zhou, 2019 [38]	Scoliosis/sagittal imbalance *n* = 130	Posterior spinal instrumentation	2 years or more	ODI VAS JOA	Retrospective LL–TK ≥ 10° LL–TK < 10°	LL-TK SVA	Postoperative LL-TK was strongly associated with patient HRQOL. Postoperative SVA is suitable for predicting the surgical outcomes for patients after correction surgery. Based on ROC curve, the cutoff value of LL-TK is ≥10° to achieve good clinical outcomes and sagittal balance.	Level IV
Sun, 2018 [48]	Spondylolisthesis Spinal stenosis Disc herniation *n* = 163	PLIA	Mean 40.6 months	Adjacent segment disease	Retrospective Multivariate analysis	PT TK	A PT of more than 24.3° and TK of more than 23.3° could be regarded as predictors for adjacent segment disease.	Level IV
Zhang, 2017 [48]	Scoliosis/sagittal imbalance *n* = 44	Posterior spinal instrumentation	Mean 2.6 years	ODI VAS JOA	Retrospective PI-LL< 10° 10° ≤ PI-LL ≤ 20° PI-LL >20°	PI-LL	The optimal PI-LL value may be achieved between 10° and 20° with good clinical outcomes.	Level IV
Sun, 2017 [37]	Scoliosis/sagittal imbalance *n* = 74	Posterior spinal instrumentation	2 years or more	Adjacent segment disease ODI VAS JOA	Retrospective PI-LL< 10° 10° ≤ PI-LL ≤ 20° PI-LL > 20°	PI-LL	The optimal PI-LL value may be achieved between 10° and 20° with excellent clinical outcomes and a lower adjacent-level disease occurrence.	Level IV
Simon, 2017 [41]	Scoliosis/sagittal imbalance *n* = 47	Posterolateral arthrodesis	Mean 6.4 years	ODI SF-36	Retrospective Univariate analysis	LL SS	Significant correlation between the SF-36 and postoperative SS and LL.	Level IV
Aoki, 2015 [49]	Spinal stenosis *n* = 52	PLIA 1 or 2 level	Mean 16.9 months	VAS Nakai scores	Retrospective PI-LL 10° or less PI-LL > 10	PI-LL	Greater PI-LL was significantly associated with worse postoperative clinical outcomes.	Level IV
Kim, 2011 [50]	Spondylolisthesis *n* = 18	Posterior interbody arthrodesis	Mean 43.1 months	ODI VAS	Retrospective Univariate analysis	PT	In patients with improved postoperative PT after arthrodesis, clinical outcomes were good regarding VAS and ODI improvement.	Level IV
Kumar, 2001 [51]	Disc herniation *n* = 83	Posterior spinal instrumentation	Mean 5 years	Adjacent segment disease	Retrospective Univariate analysis	SVA SS	Higher incidence of adjacent segment disease in patients with abnormal postoperative SS and SVA values.	Level IV
Lazennec, 2000 [52]	Spondylolisthesis Disc herniation *n* = 81	Posterior spinal instrumentation	Mean 2.8 years	Pain	Retrospective PLBP No-PLBP	PT SS	Decreased SS and PT were associated with postoperative PLBP.	Level IV

EQ-5D, EuroQol 5 dimensional questionnaire; HRQOL, health-related quality of life ; JOA, Japanese Orthopaedic Association; LL, lumbar lordosis; ODI, Oswestry Disability Index; OLIA, oblique lumbar interbody arthrodesis; PI, pelvic incidence; PLBP, persistent lower back pain; PLIA, posterior lumbar interbody arthrodesis; PT, pelvic tilt; SL, segmental lordosis; SRS-22, Scoliosis Research Society 22-item questionnaire; SS, sacral slope; SVA, sagittal vertical axis; TK, thoracic kyphosis; VAS, visual analogue scale.

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
