# Peer review of "Surgical Management of Lower Back Pain: Is Optimizing Spinopelvic Alignment Beneficial for Patient Outcomes?"

_life, 2025, doi:10.3390/life15060833_

Round 1

Reviewer 1 Report

Comments and Suggestions for Authors

Congratulations to the authors on this review. They provide a comprehensive analysis of a complex topic in spinal surgery, critically evaluating the advantages and disadvantages of various correction methods, particularly concerning clinical outcomes and overall efficacy. The study is well-structured and offers a valuable overview of the subject.

Comments on the Quality of English Language

The English should be improved to more clearly express the research.

Author Response

Thank you very much for reviewing our work and positive remarks. We have revised the use of English language and hope that it made our research more clear.  

Reviewer 2 Report

Comments and Suggestions for Authors

This review provides an illustrative overview of spinopelvic parameters and their
significance in the surgical treatment of degenerative spine disorders. While the authors have
presented an interesting and informative review, it should be noted that this is not a
systematic review. To enhance the rigor and clarity of the manuscript, the authors are
encouraged to provide a more detailed description of the search strategy employed, and
inclusion/exclusion criteria. Additionally, it would be beneficial to present the results
quantitatively (e.g., the number of studies identified) and to categorize the findings according
to their level of evidence (e.g., clinical trials, randomized controlled trials, systematic reviews,
etc.). Furthermore, a justification for the selection of the articles included in this review should
be provided to clarify the rationale behind their inclusion.

The following minor revisions are also recommended:
-Abstract, Line 21: Remove the comma after "parameters" for grammatical correctness.
-References: Ensure that reference numbers are placed immediately after the corresponding
text and before punctuation marks (e.g., [1]). This aligns with the author guidelines provided.
Currently, all references are placed after the period of the preceding sentence, which should
be corrected.
-Introduction: Consider adding a brief statement at the end of the introduction to clarify that
the anatomical aspects of spinopelvic parameters will also be reviewed. This would provide a
smoother transition into the subsequent sections.
-Section 2: The current structure introduces an additional section before the methods, which is
unconventional. It is recommended to integrate this content into the introduction or results
section.
-Figures: For improved readability, replace "See figure..." with "Fig. X" in Lines 80, 96, 106, and
112. This phrasing is more concise and aligns with standard academic writing conventions.

The authors are commended for their efforts in addressing this important topic. We wish the
authors the best of luck in finalizing their work.
Sincerely,

Author Response

Thank you for the detailed list of improvements for the article. Indeed this is not a systematic review as some of the other reviewers have remarked. Therefore we do not have a detailed Prisma chart containing the exact amount of exclusions and their reasoning. We did however revise the material and methods to display our reasoning better and provide some data of the search results. We also categorized the evidence table based on quality of evidence and study type as suggested.

  • Comma after parameters was removed.
  • References are now placed directly after the text and before punctuation marks.
  • A brief statement at the end of the introduction has been made in order to provide a smooth transition to the part discussing the spinopelvic parameters.
  • Section 2 has been added to the introduction
  • See figure X. has been changed to Fig. X.

Reviewer 3 Report

Comments and Suggestions for Authors

I read the review. It is well written and structured. However, it lacks actual interest for the readers. The introduction is very long and wordy, describing several concepts that are already well known and largely described. I suggest keeping it to a minimum and rewording unnecessary parts.

The methods section is relatively brief and scarce. The selection criteria should be expanded, and a bias assessment section added.

I suggest that authors add a strengths and limitations section at the end of the discussion paragraph.

Overall, it lacks originality and it is just a description and summary of concepts, which is better suited for a book chapter rather than a review.

Author Response

Thank you for reading our review. The introduction is indeed long and wordy, but we presume that not all readers of Life journal MDPI are spine surgeons. And even if they were spine surgeons, not all have a good understanding of the spinopelvic parameters. So we opted to include this. The material and method has been expanded. This review was written on explicit request by the Guest editor of the special issue and therefore the contents are different than that of a classic scientific paper. We aimed to illustrate our own philosophy regarding the correction of regional and global spinal imbalances and report the contemporary literature on this subject matter. Since this is not a systematic review we did not add a bias assessment section in the evidence table. However the biases are now more clearly described in the discussion.

Reviewer 4 Report

Comments and Suggestions for Authors

This is a narrative review of surgical management of low back pain focusing on whether optimizing spinopelvic alignment is beneficial for patient outcome. Overall, It is a typically general review. Some revisions are needed.

  • Narrative reviews are typically structured with an introduction, a body that discusses the key findings, and a conclusion that summarizes the main points and implications. I could not see why there is a material and methods or result section in this manuscript.
  • Figures: Within the same context, the figures should be combined into one figures with sub caption as a,b,c,... instead of numbering all sub figures.

Author Response

Thank you for reading our review.  As per request of two other reviewers we have kept the material and methods in this review and have expanded it to be more clear. We have altered the images as requested.

Round 2

Reviewer 2 Report

Comments and Suggestions for Authors

Dear authors and editors:

The authors have improved the text, and the articles they refer to are appropriate.

However, the search strategy in this review remains unclear. They mention combining 9 terms using and/or operators and excluding articles not written in English or without freely available full texts—yet it is unclear how they narrowed down from over 2,500 articles to only 19.

Additionally, the articles are still not organized according to their level of evidence. Before preparing a possible revision of the evidence grading, please review the following document:
https://pmc.ncbi.nlm.nih.gov/articles/PMC3124652/

Other minor edits:

  • Line 420: Remove the comma after "parameters".

The authors are close to a final version. Thank you for your attention to these details.

Author Response

The methodology has been expanded with reasoning for exclusion / inclusion with emphasis on study design. This remains a narrative review so the exact methodology was not documented with exact values per step of the prisma chart. The number of studies reporting clinical outcomes and linking them with spinopelvic parameters was limited and therefore less than 20 articles were selected based on over 2500 identified records. 

The articles are now organised based on level of evidence and in the conclusion a recommendation grade is given based on this level of evidence. 

Reviewer 3 Report

Comments and Suggestions for Authors

The authors responded to the reviewers' comments to the best of their ability. Overall, they improved the content of the manuscript.

Comments on the Quality of English Language

Accept after having professionally edited the English.

Author Response

The English has been revised by a native English speaker with a college degree as well as experience in medical writing. We hope in this way to have improved the manuscript.